# Bacterial Autofluorescence Digital Imaging Guides Treatment in Stage 4 Pelvic Pressure Injuries: A Preliminary Case Series

**DOI:** 10.3390/diagnostics11050839

**Published:** 2021-05-07

**Authors:** James B. Stiehl

**Affiliations:** St Mary’s Hospital, Centralia, IL 62801, USA; jim@stiehltech.com; Tel.: +1-618-780-5378

**Keywords:** pelvic pressure injuries, wound infection, autofluorescence digital imaging, decubitus, sacral wound, mechanical debridement, periprosthetic joint infection

## Abstract

Pelvic pressure injuries in long-term care facilities are at high risk for undetected infection and complications from bacterial contamination and stalling of wound healing. Contemporary wound healing methods must address this problem with mechanical debridement, wound irrigation, and balanced dressings that reduce bacterial burden to enable the normal healing process. This study evaluated the impact of bacterial autofluorescence imaging to indicate wound bacterial contamination and guide treatment for severe stage 4 pelvic pressure injuries. A handheld digital imaging system was used to perform bacterial autofluorescence imaging in darkness on five elderly, high-risk, long-term care patients with advanced stage 4 pelvic pressure injuries who were being treated for significant bacterial contamination. The prescient findings of bacterial autofluorescence imaging instigated treatment strategies and enabled close monitoring of the treatment efficacy to ameliorate the bacterial contamination. Wound sepsis recurrence, adequate wound cleansing, and diagnosis of underlying periprosthetic total joint infection were confirmed with autofluorescence imaging showing regions of high bacterial load. By providing objective information at the point of care, imaging improved understanding of the bacterial infections and guided treatment strategies.

## 1. Introduction

Pelvic pressure injuries are prevalent among older individuals in long-term care-skilled nursing facilities and contribute to significant morbidity and mortality [1]. The extent and severity can adversely impact social wellbeing, quality of life, and produce a significant burden on the health care system [2,3]. These conditions are often related to underlying neurological diseases and other comorbidities that limit mobility [4]. When left untreated, chronic wounds over the sacrum can extend to the bone and with repetitive injury, lead to large areas of involvement, including sepsis. In addition, there is a higher prevalence of multiple organism biomes, often including drug-resistant pathogens, making treatment more challenging [5]. Elderly long-term care residents have a higher prevalence of sepsis and infection with drug-resistant pathogens, making these infections all the more difficult to treat [5,6]. Sepsis is a serious condition in any patient, but particularly dangerous in those with a higher number of comorbidities. 

Clinicians managing these difficult wounds must be observant, as delayed diagnosis can lead to sepsis and death in this population. The diagnosis of high bacterial loads and sepsis can be challenging due to a decline in immune function and comorbidities that may mask clinical presentation [6,7]. Clinicians typically rely on assessment of clinical signs and symptoms combined with standard laboratory tests to determine whether infection is present. Heavy slough, malodorous smell, and chronic pain are obvious diagnostic parameters, but if there has been ongoing treatment of the wound and the patient is unable to articulate increasing symptoms, underdiagnosis of sepsis may occur [8]. To reduce this diagnostic uncertainty, clinicians need more accurate and objective methods to assess the chronic infections. 

Newer diagnostic technologies are highly relevant to clinical medicine if they demonstrate diagnostic information to inform treatment decisions that surpass current standard of care. Diagnostic point-of-care fluorescence imaging of wound bacterial burden (MolecuLight Inc., Toronto, ON, Canada) is such a technology. This point-of-care handheld non-invasive diagnostic imaging device utilizes a violet light (405 nm wavelength) in a dark environment to induce autofluorescence of bacteria. Detection of unique fluorescence signals (red and cyan) provides immediate information on the presence and location of bacteria in and around the wound. Clinical studies have reported positive predictive values >95% for these red and cyan fluorescent signals corresponding to moderate-to-heavy bacterial loads [9,10,11,12,13]. A recent multicenter study comparing clinical signs and symptoms alone to bacterial autofluorescence imaging found a 4-fold increase in sensitivity to detect wounds with >10^4^ CFU/g of bacteria with bacterial autofluorescence imaging. This bioburden hurdle was present in 131 of 350 patients where clinical signs and symptoms were negative [14]. These findings are consistent with other studies reporting improvements in detection of significant bacterial burden, and improved healing rates as a result [12,13,15,16,17]. However, the impact of this imaging technology for the diagnosis and management of sepsis in stage 4 pressure injuries has not been well described. 

This case series identifies five cases where wound sepsis in stage 4 pelvic pressure injuries was treated with a standard-of-care method, but infection was not being irradicated. We wanted to understand the infectious process and determine whether diagnostic imaging of bacterial autofluorescence could inform treatment modifications to improve our care. All patients were considered high risk, with advanced age, multiple comorbidities, and recurrent wound sepsis in several. Intention to treat focused on identifying the factors that could lead to a positive outcome. Each case revealed significant diagnostic insight gained by imaging that could alter or improve treatment imperatives. 

## 2. Materials and Methods 

### 2.1. Study Population 

An investigational review board-approved study had been created to evaluate the safety and efficacy of utilizing daily pulsatile irrigation to handle most difficult chronic infected wounds. Over two years of that study, nine patients had been treated for thirteen stage 4 pelvic pressure injuries. Of this group, five patients have been evaluated with point-of-care autofluorescence imaging to assess bacterial burden before and after the wound treatments. Each treatment included wound assessment, measurement, and daily irrigation with 3 L of low pressure using a pulsatile irrigator with an empirically measured wound surface pressure of 8 pounds per square inch. A special tip could be inserted under wound edges for cleaning wound tunneling areas. Additionally, the peri-wound area was cleansed using 2% chlorhexidine wound wipes. Participants included residents from a long-term care-skilled nursing facility with stage 4 pelvic pressure injuries. Participants were excluded from the case series if there was inability or unwillingness to consent or contraindications to routine use of pulsatile irrigation and bacterial autofluorescence imaging. Ethical approval was granted by an investigational review board (Sterling IRB #6066, Atlanta, GA, USA). Written consent was obtained from all study participants, including the use of medical images. 

### 2.2. Bacterial Autofluorescence Digital Imaging Procedure

The handheld bacterial autofluorescence imaging device was used to perform the bacterial imaging procedure at the patient’s bedside. A special plastic shroud (DarkDrape, MolecuLight Inc, Toronto, ON, Canada) was required to provide absolute darkness over the wound. (Figure 1) The autofluorescence imaging device emits a violet light that stimulates most common wound bacterial species at loads >10^4^ CFU/g to fluoresce red. This is due to molecular excitation of endogenous porphyrins, intermediates in the heme biosynthesis pathway [17]. *Pseudomonas aeruginosa* uniquely fluoresces cyan due to excitation of endogenous pyoverdine [11,18,19].

Briefly, the patient was appropriately positioned so that the wound was visible to the imaging device. Each wound was then cleansed with chlorhexidine wipes and an image was captured in standard room light. A DarkDrape was then attached to the device to create the optimal darkness required for autofluorescence imaging (Figure 1), and steps were taken to confirm darkness and position the device at the distance from the wound that is optimal for imaging. The violet light of the imaging device was then activated to capture a bacterial autofluorescence image. The clinician immediately reviewed and interpreted the image to determine whether bacteria were present in and around the wound. The presence of red or cyan fluorescence signal in the image indicated clinically significant levels of bacteria warranting treatment. The imaging device was also used to automatically calculate wound area. Calibration stickers were placed around the wound and the measurement software on the device recorded wound area (cm^2^) as well as the maximum length and width of the wound.

### 2.3. Study Group Demographics

The average age of five patients studied was 80.6 (range: 71–95). In stage 4, the pelvic wound site was the sacrum in three patients, greater trochanteric bursa in two, and other areas including three significant upper thoracic wounds in one patient and an open metatarsal phalangeal ulcer in one patient. The Charlson comorbidity index was high (>score of 5) in four cases (Avg. 6.8; range 4–9). In three patients, there had been evidence of recurrent sepsis over the long term of treatment.

## 3. Results

### 3.1. Summary of Results

In three of the five patients, red on images identified bacterial contamination greater than log 4 CFU/gram of tissue, that was found requiring treatment adjustments or considerations (Table 1; these will be further discussed as cases 1–3). In one patient, assessment of ongoing treatment revealed excellent healing and progression of the wound, with negative imaging findings, and the wound eventually closed. Another patient had multiple wounds that were possibly acute but of unknown duration. That patient had undergone multiple debridement of wounds during his hospitalization. The initial imaging scan did not reveal bacterial contamination and follow-up scans did not reveal evidence of bacterial contamination at any point. For that patient, the Charlson comorbidity index was 4.

### 3.2. Case Reports

#### 3.2.1. Case 1

A 70-year-old male had been treated for a large chronic stage 4 pelvic pressure injury over the sacrum that had been present for at least five years. He had remained bedridden during most of this time period and had multiple episodes of wound sepsis, leading to hospital readmission. Comorbidities included atrial fibrillation, stroke disease, deep venous thrombosis, and aortic stenosis, and he had been placed on a blood thinner (Xarelto)for venous prophylaxis following one of these hospitalizations.

At the five-month treatment interval, he was showing some modest bleeding with treatment and the irrigation pressure was reduced to the lower setting to ameliorate this problem. Wound slough, odor, and feeling of malaise developed and the initial assessment with autofluorescence imaging was performed. Gross wound contamination was found suggesting low-grade wound sepsis though the wound physically looked benign but clearly had active infection (see regions of red on fluorescence images in Figure 2A,B). As the patient had prior placement of a vena cava filter and he did now show active atrial fibrillation at the time, the blood thinner was discontinued and pulsatile irrigation was resumed to the high setting. This resumed the healing process and resolved the symptoms of sepsis (Figure 2C).

After an additional 8 months of treatment the wound had progressed from 35 cm^2^ to 14 cm^2^ but low-grade odor and wound slough appeared and there had been one small area where tunnelling had recured. Imaging demonstrated that complete wound cleansing of bacteria was not occurring with the daily pulsatile irrigation scheme (Figure 2D,E). A new wound swab revealed the growth of *Acinetobacter baumanii, Pseudomonas aeroginosa, and Streptococcus cloacae*. Two of these pathogens are WHO priority pathogens of concern due to high multidrug resistance risk [20]. Treatment modification was undertaken utilizing wound irrigation with a modified sodium hypochlorite antiseptic agent (Anasept). This was initially dripped onto the wound, but the manufacturer confirmed that application had been safely applied for many years with a squeeze bottle method that offered a local pressure of 8 to 12 pounds per square inch. With this information, the modified sodium hypochlorite was then run through the irrigator with the high setting and allowed to dwell for two minutes. It was then washed off with the standard 3 L saline irrigation using the pulsatile irrigator at 8 PSI. The addition was positive, allowing complete wound cleansing informed by autofluorescence imaging (Figure 2F). The patient remains stable with no sepsis recurrence and progression of wound healing.

#### 3.2.2. Case 2

A 96-year-old patient developed two stage 4 pressure injuries over the sacrum and left ischium. She was demented and poorly responsive but did not have other medical issues. The ischial wound was debrided down to the subdermal fascia and was considered clean. Three days later, the wound had dramatically changed and presented with increased erythema and heavy slough on the original wound (Figure 3A); the patient also demonstrated a low-grade fever, confusion, and severe pain suggesting sepsis. Imaging revealed red fluorescence indicative of heavy bacterial contamination of the wound bed. (Figure 3B). Immediate wound debridement uncovered a large deep abscess that expressed 20 cubic centimeters of tan purulent fluid. The patient was admitted to hospital and intravenous antibiotic treatment with vancomycin was initiated in addition to supportive measures. After three days, the patient was stabilized medically and returned to the skilled nursing facility. After 2 weeks of daily pulsatile irrigation treatment, wound assessment revealed an excellent granulating base with no slough. A digital wound area measurement (Figure 3C) was done by sounding the tunneled area with a Q-tip, estimating, and marking the limits. Digital planimetry measurement revealed 27 cm^2^ of wound area. Subsequent imaging three weeks later revealed the tunnelling diminished to 11 cm^2^, a 60% reduction (Figure 3D).

#### 3.2.3. Case 3

An elderly female, age 71, suffering from severe dementia related to stroke disease, developed pain and increased drainage from a chronic wound over her left greater trochanter (Figure 4A). The wound measured approximately 2.5 cm in diameter and was draining substantial sero-sanguinous fluid. There was mild odor, but minimal slough detected. Autofluorescence imaging revealed a bacterial contaminated fluid literally dripping from the wound (Figure 4B). The wound was palpated digitally, and the finger plunged deeper into the anterior hip cavity as the lateral hip fascia had been violated. Review of the medical record revealed a prior left hip hemi-arthroplasty that was performed five years that we were not considering before that. The immediate conclusion was chronic wound infection but draining purulence suggested deeper involvement, probably extending into the pelvis. Because of the severe mental incapacity and other medical issues, the family elected for comfort management and the patient expired two weeks later.

## 4. Discussion

This study demonstrates the advantage of utilizing point-of-care autofluorescence imaging for improving bacterial detection and optimizing treatment strategies. By any measure, the cases presented here are not usual in a typical wound practice. However, because of the magnitude of difficulty, there are lessons learned from each case report that are relevant to the practicing physician. From the first case, repeated imaging with clinical observation led to the addition of the wound wash method; modified sodium hypochlorite wash under pressure, which has enabled healing to progress. For case 2, obvious infection was suspected, and the fluorescence images revealed a compelling example of what that could look like. Case 3 represents an anecdotal observation of increased sero-sanguinous wound drainage that would not necessarily be expected from a greater trochanteric wound unless there was deeper wound involvement. Clearly, the answer was a complex periprosthetic total joint infection that had not been considered. This is an underlying fear of the treating physician that greater disease may be present, and this example offers a sign suggesting that condition. Finally, the simple measurement of the subcutaneous wound tunneling is demonstrated by using a Q-tip to sound and estimate the hole. Our clinical experience would suggest that this can occur quickly if the wound is clean and, as shown here, diminished by 60% over three weeks.

Elderly patients with pelvic pressure injuries in long-term care facilities may die from bacterial infection. Timely diagnosis and treatment of these chronic wounds is vital as reduced mobility, malnutrition, dementia, and concurrent disease can cause the wound to progress. When a patient develops sepsis, temporizing is not an option. Immediate decisions must be made with the information the clinician has at hand. The more objective information available, the more clinically sound the decisions can be. In the present series, a clear advantage was identified that allowed clinicians to confirm pathological bacteria loads at >10^4^ CFU/g. This threshold is considered the diagnostic hurdle for satisfactory wound healing and defined as the minimum bacterial bioburden that will develop infection if left untreated [21,22].

This case report relied on the previously published clinical study by Rennie et.al. that analyzed the performance of the MolecuLight system for measuring a significant clinical wound infection at log4 to log 5, CFU/gm tissue [12] The Rennie study attempted to define the bacteria load by assessing surface and surface punch biopsies guided by autofluorescence imaging. Currettage scrapings underwent semi-quantitative culture analysis while the biopsies utilized 16S quantitative polymerase chain reaction to create a value for total bacterial load in CFU/gram. When red fluorescence was present in the biopsy specimen, the positive predictive value was 100%. The 16S gene targeted quantitative polymerase chain reaction yielded growth from 10^4^ to 10^8^ CFU/g. This study confirmed the ability of the autofluoresence imaging to guide biopsy and treatment when bacterial growth exceeds 10^4^ CFU/g.

Wound imaging for fluorescence provided additional information on the location of bacterial burden, with large loads of bacteria identified on the peri-wound area. This is not surprising, and is in line with findings from Bay et al., who identified increased aggregates of bacteria in these peripheral wound edges [22]. Le et al. and Raizman et al. [14,16] each reported higher bacterial loads from wound samples taken from regions of red or cyan fluorescence in the peri-wound. The thoughtful conclusion is that wound cleansing must be as extensive as needed for improved wound care and simply judging the appearance of the wound may not be enough.

With any new technology, there can be a learning curve. To obtain adequate autofluorescence images, the black plastic shroud (DarkDrape) is essential in rooms that cannot be fully darkened. For the planimetry measurement images, careful attention is needed to place the calibration stickers reasonably close together on a parallel surface of skin. Additionally, the ideal focal distance is determined with an indicator light for cue and a digital image grid that targets the desired area of study. One final advantage is the ability of the device to store unlimited numbers of images that can be valuable for clinical study, and its capability for telemedicine based wound care, with strong implications for expedited treatment.

The primary limitation of this study is the lack of clinical follow up but the cases come from a pre-hypothesis IRB study where the study design was primarily intention to treat. We wanted to study very difficult cases that had failed prior treatment and chronic wounds that had stalled after months of treatment. Just one of these cases had healed the wound during this study and two had died, not completing treatment. The point of this study is the binary answers offered by the images; either the wound is infected or the infection is under control. The cases chosen clearly reflect this objective.

Bacterial autofluorescence imaging is a powerful new technology that guides clinicians to better treatment methods. Studies indicate that without this imaging information, treatment plans fail to adequately address bacterial burden in 70% of wounds [23]. In the difficult cases described here, we utilized autofluorescence imaging to confirm the efficacy of daily low pressure pulsatile irrigation to remove most if not all imaged bacteria at each treatment. Recalcitrant bacterial aggregates could be addressed by repeated rounds of imaging informed debridement and improved selection of cleanser when images have demonstrated ineffectiveness of the initial selection.

## 5. Conclusions

Bacteria-induced infection in chronic wounds continues to be a pervasive challenge that comes at a great cost to the health care system, and to patients and their families. Pelvic pressure injuries are notorious for unrecognized and deep tunneled infection that may lead to systemic sepsis. Utilizing bacterial autofluorescence imaging, we immediately identified active infections, that we believe could have led to life threatening sepsis if not adequately addressed. It is projected that bacterial autofluorescence imaging will become a standard-of-care diagnostic imaging procedure.

## Figures and Tables

**Figure 1 diagnostics-11-00839-f001:**
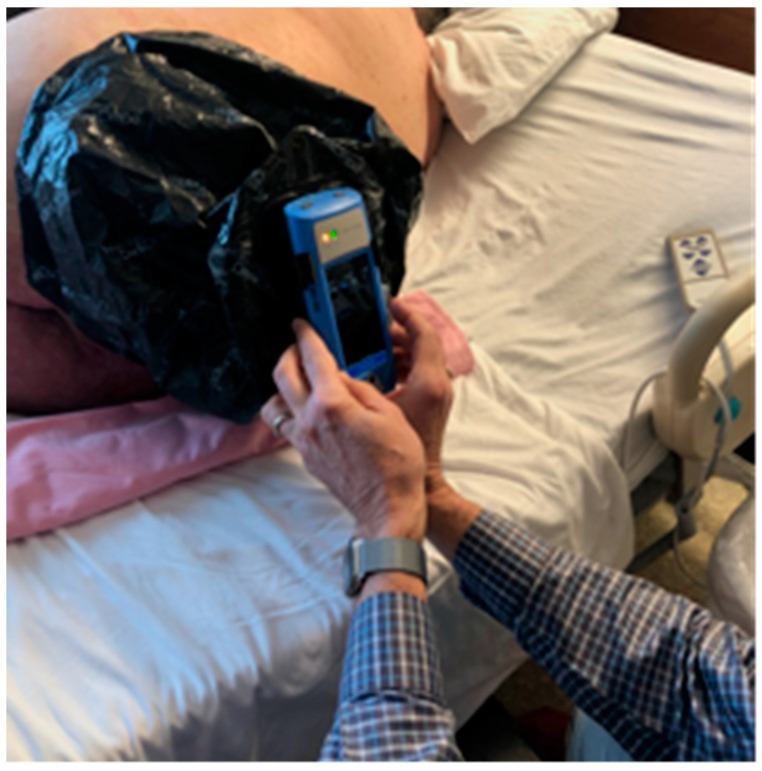
Bacterial autofluorescence image capture with a DarkDrape. A dark environment is required to capture interpretable fluorescence images. A DarkDrape is used when room lights cannot be turned off or other environmental conditions cause light contamination.

**Figure 2 diagnostics-11-00839-f002:**
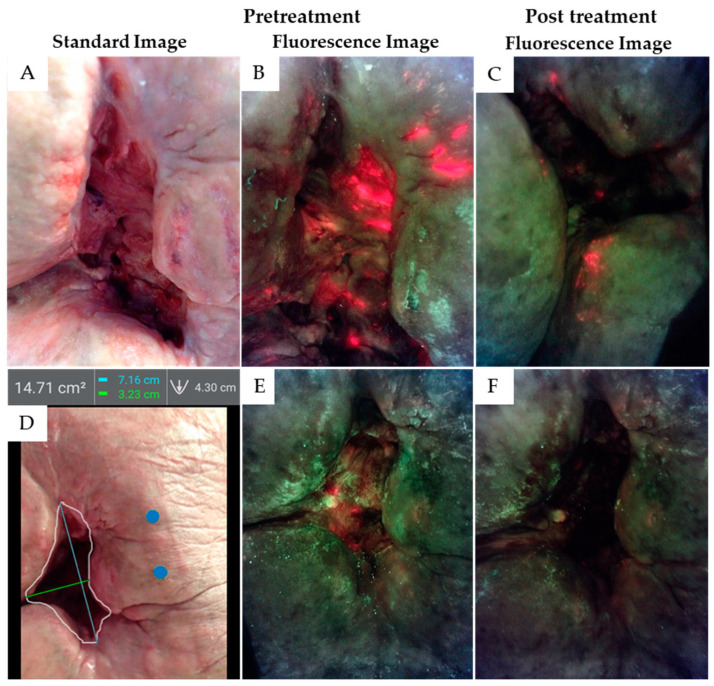
**Case 1**. (**A**) This stage 4 pelvic pressure injury (35 cm^2^) over the sacrum had been treated with jet lavage irrigation for approximately 5 months. This patient developed signs of septicemia, though this wound has modest slough and does not appear grossly infected. (**B**) Autofluorescence bacterial imaging with a DarkDrape revealed extensive bacterial infestation demonstrated by red fluorescence observed in the wound bed and peri-wound area. After treatment, red bacterial autofluorescence was reduced but still present (**C**). Eight months later, a 25% reduction in wound size (**D**) was observed (blue dots calibrate planimetry measurements), but some red bacterial fluorescence persisted (**E**), prompting additional treatment. After treatment using pulsatile irrigation and antiseptic under pressure, the wound was re-imaged and bacterial autofluorescence was no longer detected (**F**).

**Figure 3 diagnostics-11-00839-f003:**
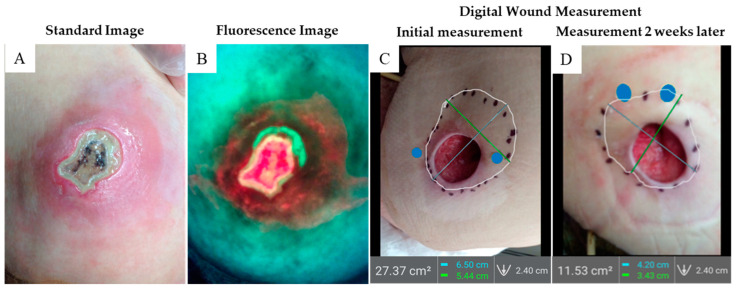
**Case 2.** This stage 4 pelvic pressure injury (6 cm^2^) over the left ischium had developed necrotic tissue over the subdermal fascia and has a broad area of erythema in surrounding skin. (**A**) Three days before, wound appearance had been benign after debridement. (**B**) Autofluorescence reveals marked bacterial infestation in the wound bed and also alerted the clinician to extensive diffuse subcutaneous bacterial burden. (**C**) A digital wound measurement was obtained with the bacterial imaging device by placing 2 blue WoundStickers on either side of the wound assessing the tunneled area to 27 cm^2^ (**D**). Three weeks later, digital wound measurement showed a dramatic reducing in wound size down to 11.5 cm^2^, a 60% reduction.

**Figure 4 diagnostics-11-00839-f004:**
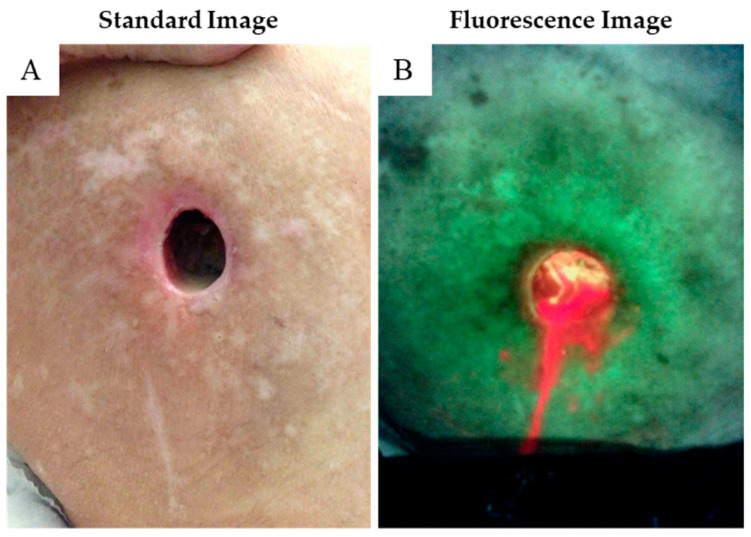
**Case 3.** (**A**) This wound over the right greater trochanter measures 1.5 cm and has clear serous drainage. (**B**) Drainage from the wound shows red autofluorescence of bacterial infestation, not recognized by physical examination alone. This proved to be a periprosthetic total joint infection.

**Table 1 diagnostics-11-00839-t001:** Summary of cases and imaging findings.

Case	Age	Charlson Index ^1^	Sepsis Diagnosis	Autofluorescence Imaging Findings	Impact of Imaging
1	71	7	Yes, informed by images	Gross wound contamination (red fluorescence) was found suggesting low-grade wound sepsis, despite the wound physically looking benign	(1) Demonstrated objectively the insufficiency of the prescribed cleansing regime, (2) led to a switch to cleanser with irrigation and enabled immediate evaluation of its efficacy, (3) prompted collection of a wound culture
2	95	6	Yes, informed by images	Widespread red fluorescence indicative of heavy bacterial contamination of the wound bed	Severity of imaging findings prompted immediate wound debridement, which uncovered a large, deep abscess and led to antibiotic prescription
3	71	8	Yes, informed by images	Red fluorescent pus readily observed on images of sero-sanguinous wound fluid	Large wound fluid amount prompted further clinical investigation revealing probable periprosthetic total join infection
4	76	9	no	Initial images and follow-up scans were negative for bacterial loads of concern.	Indicated that advanced treatments were unnecessary; wound healed without incident
5	88	4	no	Across multiple wounds, initial images and follow-up scans were negative for bacterial loads of concern.	Indicated that advanced treatments were unnecessary; wound continues to progress well

^1^ Charlson comorbidity index; score > 5 considered high.

## Data Availability

Not appliable.

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
