# Peer review of "Bacterial Autofluorescence Digital Imaging Guides Treatment in Stage 4 Pelvic Pressure Injuries: A Preliminary Case Series"

_diagnostics, 2021, doi:10.3390/diagnostics11050839_

Round 1

Reviewer 1 Report

In this paper, the impact of bacterial autofluorescence imaging to indicate bacterial contamination in wounds for severe stage 4 pelvic pressure injuries was evaluated.

In my opinion, the study was systematic and organized. The experiments were carefully executed and the discussion was coherent. Results obtained were interesting and were consistent with the conclusions.

In summary, the paper was put together well and contained interesting new data hence it will be of interest to other researchers working in this area as well as to the readership of this journal.

Author Response

Thanks to this reviewer!  I am not finding any corrections to make and may assume that the reviewer is satisfied.

Please correct me if I am wrong.

Reviewer 2 Report

The authors try to apply bacterial autofluorescence imaging instigated to facilitate efficacy monitoring after clinical treatment. However, there is some concern need to be addressed.

 Major concerned

  1. The authors just reference the previous study results but not demonstrated them in their system to prove that they can achieve the sensitivity as >104 CFU/g of bacteria.
  2. Just five patients have been included in this study, and only three of them have finished the whole evaluation process.

Author Response

In response to reviewer #2, the request, I think is to describe the medical evidence that supports the current work.  That comes primarily from the Jones, et. al. article and I add a new paragraph summarizing that evidence.

The second question, I think is dealing with why the study was done, and I add a new paragraph.  This is basically an intention to treat format, and the interesting part of each case is not the clinical outcome but the point of fact decision making that was apparent with the addition of the imaging. I hope my description solves the issue.

This case report relied on the previously published clinical study by Jones, et.al. that analyzed the performance of the MolecuLight system for measuring a significant clinical wound infection at log4 to log 5, CFU/gm tissue. The Jones study attempted to define the bacteria load by assessing surface and surface punch biopsies guided by autofluorescence imaging. Currettage scrapings underwent semi-quantitive culture analysis while the biopsies utilized 16S quantitative polymerase chain reaction to create a value for total bacterial load in CFU/gram. When red fluorescence was present in the biopsy specimen, the positive predictive value was 100%. The 16S gene targeted quantitative polymerase chain reaction yielded growth from 104 CFU/g to 108CFU/g. This study confirmed the ability of the autofluoresence imaging to guide biopsy and treatment when bacterial growth exceeds 104 CFU/g. 

The primary limitation of this study is the lack of clinical follow-up but the cases come from a pre-hypothesis IRB study where the study design was primarily intention to treat. We wanted to study very difficult cases that had failed prior treatment and chronic wounds that had stalled after months of treatment. Just one of these cases had healed the wound during this study and two had died, not completing treatment. The point of this study is the binary answers offered by the images; either the wound is infected or the infection is under control. The cases chosen clearly reflect this objective.